# Prediction of Radiation-Induced Hypothyroidism Using Radiomic Data Analysis Does Not Show Superiority over Standard Normal Tissue Complication Models

**DOI:** 10.3390/cancers13215584

**Published:** 2021-11-08

**Authors:** Urszula Smyczynska, Szymon Grabia, Zuzanna Nowicka, Anna Papis-Ubych, Robert Bibik, Tomasz Latusek, Tomasz Rutkowski, Jacek Fijuth, Wojciech Fendler, Bartlomiej Tomasik

**Affiliations:** 1Department of Biostatistics and Translational Medicine, Medical University of Lodz, 92-215 Lodz, Poland; urszula.smyczynska@umed.lodz.pl (U.S.); szymon.grabia@umed.lodz.pl (S.G.); zuzanna.nowicka@umed.lodz.pl (Z.N.); bartlomiej.tomasik@umed.lodz.pl (B.T.); 2Department of Radiotherapy, N. Copernicus Memorial Regional Specialist Hospital, 93-513 Lodz, Poland; anna_papis@interia.pl (A.P.-U.); jacek.fijuth@umed.lodz.pl (J.F.); 3Department of Radiation Oncology, Oncology Center of Radom, 26-600 Radom, Poland; rmbibik@wp.pl; 4Radiotherapy Department, Maria Sklodowska-Curie National Research Institute of Oncology (MSCNRIO)—Branch in Gliwice, 44-101 Gliwice, Poland; tomasz.latusek@io.gliwice.pl; 5I Radiation and Clinical Oncology Department, Maria Sklodowska-Curie National Research Institute of Oncology (MSCNRIO)—Branch in Gliwice, 44-101 Gliwice, Poland; Tomasz.Rutkowski@io.gliwice.pl; 6Department of Radiotherapy, Chair of Oncology, Medical University of Lodz, 93-509 Lodz, Poland; 7Department of Radiation Oncology, Dana-Farber Cancer Institute, Boston, MA 02115, USA

**Keywords:** radiomics, radiation-induced hypothyroidism, NTCP, head, neck cancer

## Abstract

**Simple Summary:**

Radiation-induced hypothyroidism (RIHT) commonly develops in cancer survivors that receive radiation therapy for cancers in the head and neck region. The state-of-art normal tissue complication probability (NTCP) models perform satisfactorily; however, they do not use the whole spectrum of information that can be obtained from imaging techniques. The radiomic approach offers the ability to efficiently mine features, which are imperceptible to the human eye, but may provide crucial data about the patient’s condition. We gathered CT images and clinical data from 98 patients undergoing radiotherapy for head and neck cancers, 27 of whom later developed RIHT. For them, we created machine-learning models to predict RIHT using automatically extracted radiomic features and appropriate clinical and dosimetric parameters. We also validated the well-established external state-of-art NTCP models on our datasets and observed that our radiomic-based models performed very similarly to them. This shows that automated tools may perform as well as the current standard but can be theoretically applied faster and be implemented into existing imaging software used when planning radiotherapy.

**Abstract:**

State-of-art normal tissue complication probability (NTCP) models do not take into account more complex individual anatomical variations, which can be objectively quantitated and compared in radiomic analysis. The goal of this project was development of radiomic NTCP model for radiation-induced hypothyroidism (RIHT) using imaging biomarkers (radiomics). We gathered CT images and clinical data from 98 patients, who underwent intensity-modulated radiation therapy (IMRT) for head and neck cancers with a planned total dose of 70.0 Gy (33–35 fractions). During the 28-month (median) follow-up 27 patients (28%) developed RIHT. For each patient, we extracted 1316 radiomic features from original and transformed images using manually contoured thyroid masks. Creating models based on clinical, radiomic features or a combination thereof, we considered 3 variants of data preprocessing. Based on their performance metrics (sensitivity, specificity), we picked best models for each variant ((0.8, 0.96), (0.9, 0.93), (0.9, 0.89) variant-wise) and compared them with external NTCP models ((0.82, 0.88), (0.82, 0.88), (0.76, 0.91)). We showed that radiomic-based models did not outperform state-of-art NTCP models (*p* > 0.05). The potential benefit of radiomic-based approach is that it is dose-independent, and models can be used prior to treatment planning allowing faster selection of susceptible population.

## 1. Introduction

Patients with head and neck cancer (HNC) treated with radiation therapy (RT) may experience multiple adverse normal tissue effects [1], including hypothyroidism. Radiation-induced hypothyroidism (RIHT) has been reported in 25–65% of patients and typically develops during the first 2 years from RT completion. Since the symptoms of RIHT are non-specific and insufficient levels of thyroid hormones not only negatively impact patients’ quality of life [2], but also mortality [3,4], adequate identification of patients at risk of this effect is of paramount importance [5,6,7].

Clinical and dosimetric parameters are typically used in normal tissue complication probability (NTCP) models to predict RIHT [1]. Recently, we [7] and others [6] have applied published NTCP models in homogenous cohorts of patients with oropharyngeal cancer (OPC). We concluded that two models based on thyroid mean dose and volume, published by Rønjom et al. [8] and by Boomsma et al. [9], performed best in terms of accuracy (84 to 87%), highlighting the feasibility of dose-response models to predict RIHT and their potential utility in the clinical setting.

Radiomics is a relatively new discipline that aims at deriving biomarkers from medical images [10]. These biomarkers are features describing shape, intensity or texture of specific region(s) of interest (ROI), typical an organ or lesion. Radiomics emerged in the field of cancer studies where large number of medical images are generated even by routine diagnostic process. In this context, radiomics can supplement standard radiologic assessment with localized, quantitative information on interesting structures with low cost and without any additional burden or discomfort to patients. In recent years, radiomics researchers community took considerable efforts to standardize methodology aiming at translatability of radiomic studies that resulted in publication of the first reference manual for image biomarker studies [11].

Radiomic features were shown to reflect cancer biology in terms of histologic type [12], overall and progression-free survival [12,13,14], probability of recurrence [15], activity of biological pathways [12,16], human papillomavirus infection [17], probability lymph node [18] or distant metastases [19], CD8 cells infiltration [20]. Some studies extended radiomic analysis outside tumor volume, including also peritumoral regions for prediction of patients’ survival [21] or adjuvant chemotherapy [22]. Similarly, lymph nodes were sometimes considered to be ROI in studies that aimed at identifying lymph nodes metastases [23,24]. Some radiomic NTCP models based on computed tomography (CT) were developed, for instance for prediction of xerostomia after head and neck cancer radiotherapy [25,26], radiation-induced pneumonitis [27]. Alternatively, radiomic features for radiation-induced NTCP models could be calculated in 3D dose distribution that was successfully done for cervical cancer [28], prostate cancer [29] and lung cancer [30]. Given the very good performance of some NTCP models in predicting RIHT in patients with OPC, the question whether there would be an added benefit of an in depth radiomic analysis was unresolved and thus we decided to apply CT-based radiomics in this group of patients and compare predictive performance with the available models based on dosimetric and clinical features.

## 2. Materials and Methods

### 2.1. Patients and CT Images

The studied cohort was a subgroup of patients included in previous study on validation of NTCP models for radiation-induced hypothyroidism by Nowicka et al. [7], recruited between 1 May 2016 and 31 December 2018 and followed up until February 2020. In addition to already collected clinical data, for this study we retrieved CT images from radiotherapy planning with thyroid glands retrospectively contoured by two experienced radiation oncologists according to the guidelines for organs at risk (OARs), as described previously. Patients for whom the CT image was unavailable, thyroid contour was missing or thyroid region contained CT artifacts, were excluded. Finally, among 108 patients recruited in 3 centers 98 had complete data and were included in this study (38 from center A-Copernicus Regional Specialist Hospital in Lodz, Poland; 12 from center B-Radom Oncology Center and Maria Sklodowska-Curie National Research Institute of Oncology, Radom, Poland and 48 from center C-Radiotherapy Department, Maria Sklodowska-Curie National Research Institute of Oncology —branch in Gliwice, Poland); selection of cases is summarized in Figure 1A.

All patients underwent intensity-modulated radiation therapy (IMRT) of OPC before which their thyroid function was normal. After RT, they were monitored for a median of 28 months on average (median). During this follow-up period, 27 patients developed hypothyroidism. To avoid bias, laboratory assessment, contouring, outcome assessments, statistical and radiomic analysis were performed by independent researchers. Details of treatment protocol were described previously [7].

For all 98 patients, CT images paired with RT plans and anatomical structure contouring files were retrieved from PACS of treating centers. All images were stored in DICOM format in CT and RTSTRUCT modalities; however, some differences in acquisition protocol were identified. First, each center used a different CT scanner (center A: Somatom Sensation Open, Siemens; center B: Optima CT580 RT, GE Healthcare; center C: SOMATOM Definition AS and Somatom Sensation Open, Siemens). In each center, slice thickness and pixel spacing were selected by radiotherapist and radiologist so that images were sufficient for treatment planning. Summary of selected settings in each center is presented in Table 1 and anonymized raw patients data are in Appendix A.

The study was approved by the Bioethics Committee of the Medical University of Lodz (KE/7/10, RNN/65/18).

### 2.2. Image Preprocessing and Radiomic Features Calculation

Image processing and radiomic feature extraction were performed with PyRadiomics v3.0 [31]. Due to diversity of pixel spacing and slice thicknesses, all images and thyroid masks (generated from contours using dcmrtstruct2nii library [32]) were resampled to 1 × 1 × 1 mm^3^ isotropic voxels. Default PyRadiomics interpolators were used in resampling: nearest neighbor interpolation for binary thyroid mask and B-spline interpolation for CT image. At the same time, both image and mask were cropped to thyroid bounding box with 10 mm padding added to ROI bounding box at each side; the operation is done by default by PyRadiomics during resampling, but we increased default padding size from 5 to 10 voxels. Then, radiomic features were calculated that in the applied version of the library included: 14 shape features, 18 first order statistics, 24 gray level cooccurrence matrix-based (GLCM) features, 16 gray level run length matrix-based (GLRLM) features, 16 gray level size zone matrix-based (GLSZM) features, 5 neighborhood gray tone difference matrix-based (NGTDM) features, 14 gray level dependence matrix-based (GLDM) features (full list of features in Appendix A). Radiomic features were extracted from original images and from their filtered versions, applying all filters available in PyRadiomics feature extractor: square, square root, logarithm, gradient, exponential and 8 wavelet decomposition filters. Default values of PyRadiomics feature extractor settings were used, including filtration parameters. Resegmentation was not applied. In total, 1316 features were calculated for each image (14 shapes features extracted from image mask, 93 intensity-based features for original images and 13 filtered ones).

The processing of images and NTCP model derivation is summarized in Figure 1B, while raw radiomic features values are reported in Appendix A.

### 2.3. Stability Assessment

The purpose of stability analysis was to identify features that were unaffected (or affected only slightly) by minor differences in contouring of thyroid glands. As the segmentations by multiple radiotherapists or radiologists were not available, we decided to perform a simulation study. We modelled inaccuracies of contouring as affine deformations of thyroid mask, consisting of:3 translations by up to 1 mm in either direction along each of the 3 main axes,3 rotations by up to 2° in either direction around each of the 3 main axes,3 zooms by up to 2% of either dimension along each of the 3 main axes.

We generated 100 such transformations by randomly choosing order and parameters of the above 9 operations. Then, we applied every transformation to all thyroid masks in our image data set and used the transformed masks to calculate new sets of radiomic features. These transformed masks were used solely for the purpose of stability assessment, while for model development we calculated features using original manual contours of thyroid glands.

Inter-class correlation coefficient (ICC) served as a measure of agreement between original features (calculated using unchanged mask) and those calculated using each of 100 transformed masks. Later, 100 ICC values obtained for different transformed masks were averaged to give single measure of stability for each feature. Stability was classified as excellent when mean ICC exceeded 0.9, good (ICC between 0.75 and 0.9), moderate (ICC between 0.5 and 0.75) or poor (ICC not exceeding 0.5).

### 2.4. Feature Preprocessing and Splitting Data Set

Raw feature values were first scaled to a [−1,1] range, then subjected to Yeo–Johnson transformation [33] and scaled to [−1,1] range once again to facilitate machine-learning model derivation. First scaling was performed to avoid exponentiation (included in Yeo–Johnson transformation) of raw radiomic features with high absolute values, such as energy that easily reached values at the order of 10^10^. Second scaling aimed at facilitating model training by limiting the range of values generated by Yeo–Johnson transformation. As average values and distributions of features values varied across the samples (Appendix A) we normalized them dividing each value by sample mean. While performing normalization, we followed the reasoning common for other high-throughput studies (transcriptomics, proteomics) that usually only minority of features should be expected to differ between conditions, thus the distribution of all feature values should not differ significantly between samples.

We observed that even after normalization, we could still observe batch effect related to clinics that performed CT scans (Appendix A). Thus, to account for this and check the impact of batch on NTCP model performance, we decided to implement different variants of data preprocessing and splitting:Variant I without batch effect correction, later referred to as Variant Ia: center A as training set, centers B and C as validation set as shown in Figure 1A.Variant I with batch effect removed by ComBat [34], referred to also as Variant Ib: center A as training set, centers B and C as validation set.Variant II: no batch effect removal, but data from three centers were joined and subsequently divided into training and validation sets (Figure 1A). Training set contained 38 patients including 10 RITH to match the number of patients and proportion of RITH cases of center A data set so that both variants of splitting data are as comparable as possible.

### 2.5. Feature Filtration

Due to the large number of features, we decided to filter them before derivation of NTCP models. First, only features with excellent or good stability were considered to be candidates for NTCP model predictors. We used ICC classes from center A for filtration in Variant I and, analogously ICC classes from training set for Variant II.

Then, to retain only features that actually differentiate patient with and without RIHT in follow-up, t-tests were performed to compare values of each stable feature between these group. Benjamini–Hochberg correction was applied to p-values to control false discovery rate (FDR) in this large set of comparisons. Next, hierarchical clustering of features was performed with 1 − r (correlation coefficient) as distance measure and average linkage to identify groups of highly correlated features. These groups of features were extracted by setting a threshold of 0.3 to distances in the dendrogram. From each such group we retained a single feature with lowest FDR-corrected p-value (later referred to as FDR) on the condition that this FDR did not exceed 0.1. Additionally, we kept all features with FDR < 0.01 even if they were correlated with others. This shortened list of features was the basis for models’ training that included final model-based feature selection.

### 2.6. Model Training and Evaluation

Model training was performed with the use of scikit-learn Python library [35]. Model architectures considered in our analysis included: logistic regression, multilayer perceptron (MLP), k nearest neighbors classifier, support vector classifier, decision tree, random forest, AdaBoost classifier, Gaussian process classifier, Gaussian Naive Bayes classifier, Quadratic discriminant analysis. Whenever possible (in logistic regression, support vector classifiers, decision trees and random forest) two methods of weighting cases during training were applied: (1) equal weights of all cases (2) weights inversely proportional to classes’ frequencies to account for imbalanced data set with less than 1/3 of cases in RIHT group. Full list of tested model parameters is collected in Appendix A.

Models were derived using three feature sets: (1) only clinical and dosimetric features (clinical model), (2) only radiomic features (radiomic model), (3) radiomic, clinical and dosimetric features (radiomic+clinical model). Final set of input features for each model was determined by forward method of Sequential Feature Selector from mlxtend library [36]. Feature selector used area under the ROC curve (AUC) as a measure of models’ performance and was allowed to choose from 2 to 5 features.

All models were first derived using training data and then their performance was tested on validation data. AUC, accuracy, sensitivity, specificity and F-score were calculated for each model. The analysis was performed for Variants Ia, Ib and II of clinical and radiomic features derivation. Models with the highest F-score were selected for each variant of analysis and input feature set. Finally, ensembles of best clinical and radiomic models were considered that combined them in following ways:logical conjunction (AND): positive prediction only when both models predicted RIHT,logical disjunction (OR): positive prediction when any of the two models predicted RIHT,averaged probability (PROBA): probability (raw output) of two models were averaged and new decision threshold selected using ROC curve for training set.

## 3. Results

### 3.1. Feature Stability Analysis

Results of feature stability analysis differed slightly between variant I and II. In variant I, where all images were acquired in the same center, we identified more features as highly stable (ICC > 0.9) than in variant II (721, Figure 2A vs. 621 features, Figure 2B). In line with this observation, lowest stability class was assigned to 211 in variant II and only to 138 features in variant I (stability analysis was performed before batch correction, thus at this stage variant Ia and Ib are not distinguished). Independently of the variant, all shape features were excellently stable even though our test consisted essentially in disturbing thyroid mask shape. Excellent or good stability was observed for majority of first order, GLCM, GLRLM, and GLDM features, while for GLSZM and NGTDM the fraction of unstable features was higher. In majority, stable features overlapped between variants I and II (Figure 2C). Stability of features depended also on the applied filter (Figure 2D), with exponential and gradient filter ensuring on average higher stability than square, square root, logarithm and some wavelet filters.

In further analysis, we included all features with excellent and good stability: 926 in variant I and 869 in variant II. Stability classes for all features are reported in Appendix A.

### 3.2. Feature Processing and Filtration

Before filtration, features were scaled, normalized (Appendix A) and in variant Ib batch effect was corrected (Appendix A). First stage of filtration consisted of exclusion of features that in univariate analysis did not differentiate patients who did or did not develop RIHT (detailed results of this analysis in Appendix A). It resulted in selection of 165 radiomic features in variant I and 166 in variant II, among which 153 were common to both variants (Figure 3A). Having a limited number of patients, we decided to reduce these numbers further before model development by elimination of highly correlated features. Selection of representatives from each group of such related variables left 67 features in variant Ia, 68 in variant Ib and 66 in variant II (lists of features in Appendix A). Again, the overlap between variants, consisting of 61 radiomic features, greatly outnumbered distinct variables for each variant (Figure 3C). At both stages of filtration, features from original image and the one with exponential filter applied were preferred over other groups (Figure 3B,D).

### 3.3. Models

The training of all the considered models on the reduced set of features and their validation resulted in selection of top 9 models, one for each variant and a feature set. Their names, along with the features they were built upon, are presented in Table 2, while details of all analyzed models can be found in Appendix A.

For all the variants, when considering models derived only with clinical features, Gaussian process classifier was chosen. Likewise, the same set of features was selected, i.e., mean of thyroid dose, median of thyroid dose and volume of the thyroid. For radiomic and clinical+radiomic feature sets, the selected models were logistic regression without regularization, with equal class weights and excluded intercept for variant Ia and multilayer perceptron with 4 and 2 hidden neurons for variants Ib and II, respectively. For those feature sets, selected features were more diverse.

In all variants of analysis, radiomic models performed similarly to clinical models (Figure 4A–C); comparisons of ROC AUC by DeLong test never showed superiority of models using radiomic features vs clinical/dosimetric model (Table 3). Slight improvement of radiomic models is observed after batch effect correction (variant Ib). The statistical measures (sensitivity, specificity, accuracy and f-score) for each of the selected models and the ensembles of clinical and radiomic models were calculated (Figure 4D–I). Radiomic+clinical model was not included in ensembles, because it contained a very similar feature set to radiomic and clinical models taken together. Based on those measures, we picked clinical, radiomic and OR models for the comparison with the external NTCP models for radiation-induced hypothyroidism [8,9,37,38,39].

Compared with previous examples (Figure 5), our models tend to be slightly less sensitive, but more specific and accurate. Models by Cella et al. and Vogelious et al. significantly overestimate risk of RIHT, declaring all or almost all patients as having high risk of this complication. The model by Ronjom et al. seemed comparable to our models.

Radiomic features included in the final models came from original, logarithm, exponential and wavelet (LLL, HHH) images. Majority of these features measured nonuniformity of the thyroid region (e.g., coarseness, zone percentage) and were higher in patients who developed RIHT (Figure 6). The only feature lower in these patients was least axis length of thyroid (Appendix A).

## 4. Discussion

Here, using data from 98 patients with OPC treated with definite RT, we identified radiomic features predicting the occurrence of RIHT in 2-year follow-up and contrasted our radiomic-based model with published NTCP models based on clinical and dosimetric parameters. Our model performed comparably to models published by Rønjom et al. [8] and by Boomsma et al. [9] and better than three other external models [37,38,39]. Since predictions based on CT-derived radiomic features are agnostic to dose distribution, they reflect the baseline susceptibility of individual thyroid glands to damage inflicted by radiation doses typical for head and neck cancer RT. They may be therefore leveraged to identify patients in whom close attention should be paid to minimize radiation to the thyroid and the risk of RIHT; alternatively, dosimetric/clinical and radiomic models can be combined to improve prediction accuracy.

Most features in our radiomic model were calculated in filtered CT images and described nonuniformity or coarseness of thyroid region, sometimes emphasizing low gray levels. Invariably higher values of these features, indicating greater non-homogeneity, especially of darker (lower Hounsfield units) regions of thyroid, were characteristic for patients who later developed RIHT.

It must be noted that several articles reported that attenuation and size of an otherwise normal-appearing thyroid might serve as an imaging biomarker for hypothyroidism. Noteworthy, this phenomenon was described for both diagnostic [40] and low dose CT [41].Approximately 25% of the body iodine is accumulated in the thyroid gland [42]. This results in hyperdense appearance on the CT scan. Older works postulated that decreased density may be related to disturbances in iodine accumulation and/or processing e.g., inflammation [43,44]. Providing biological rationale behind the signal from radiomic features is beyond the scope of our study; however, we can speculate that our radiomic pipeline might detect even more subtle differences in the signal from thyroid gland. Such disturbances may reflect subclinical thyroid insufficiency which makes the gland more susceptible and/or less prone to regeneration.

Differences in performance between our models and external ones as well as those between different external models likely stem from differences in patient cohort characteristics and chosen treatment protocols, which has already been discussed in detail in a previous study by Nowicka et al. [7] that used data from the same cohort of patients.

An important contribution of our study is analytical pipeline that adds to the recommendations from first reference manual for image biomarker studies [11] and Radiomic Quality Score (RQS) [45]. Addressing criteria from RQS, we reported imaging protocols from participating institutions and verified the stability of features by segmentation perturbation, reducing the number of features and applying multiple comparisons correction. Our simulation study showed that some features are unstable with respect to inaccuracies in thyroid contouring; however, we identified a subset of relatively stable features, corresponding with the results of study on robustness of radiomic features by Zwanenburg et al. [46]. Then, we scaled and normalized the features and performed batch effect correction in one variant of the analysis. This pipeline may be reused and extended for similar projects requiring combining data from many institutions or imaging machines.

We included non-radiomic patient characteristics in the study, reported models’ quality statistics and validated models. Patients were recruited prospectively in 3 clinics for the study by Nowicka et al. with the same endpoint of RIHT [7]; however, radiomic analysis was included in the protocol at the later stage. In the place of validation against “golden standard”, which is not established for prediction of RIHT, we compared our models with those by other authors. As reported by reviews of radiomic studies [47,48], full adherence to the guidelines is rarely achieved; however even partial compliance improves the quality of research.” A TRIPOD guidelines checklist that describes the critical aspects of our work has been provided in the Appendix A.

Our study has several limitations inherent to radiomic studies. The observed batch effects of radiomic features related to oncologic centers (possibly due to use of different CT machines or their settings) may be, for research purposes, solved by batch effect removal tools developed for other high-throughput studies [34,49]. However, such procedure complicates the translation of results and has not been extensively validated in terms of preserving data integrity. Furthermore, our data set has a relatively small sample size with respect to number of features and the developed models require further validation before they can be applied in the clinical setting. Although the validation groups were sufficiently numbered to detect statistically significant differences exceeding 13% of accuracy it would mandate a larger study to confirm true non-interiority or equivalence of the radiomic and NTCP models, but owing to the presented results planning such an endeavor is possible. Estimating the sample size, if we were to achieve significant difference between AUC of clinical and radiomic models, keeping the current ratio of patient with the complications, we would have to collect data from at least 179 patients who developed RIHT after radiotherapy and from 500 patients, who did not.

An additional benefit of validation in external cohort, ideally from different population, is verification whether the models are not overfitted to our study population. Testing of models developed using data from center A (variant I) on data from center B and C did not indicate significant overfitting. However, this might be also the result of similarity of studied patient groups that all recruited from Polish population that is quite uniform ethnically.

## 5. Conclusions

Radiomic models reliant on CT scans showed a similar or better ability to predict RIHT in OPC patients compared with the best currently used NTCP models. Additionally, radiomic models are independent from treatment planning and readily deployable across different imaging data.

## Figures and Tables

**Figure 1 cancers-13-05584-f001:**
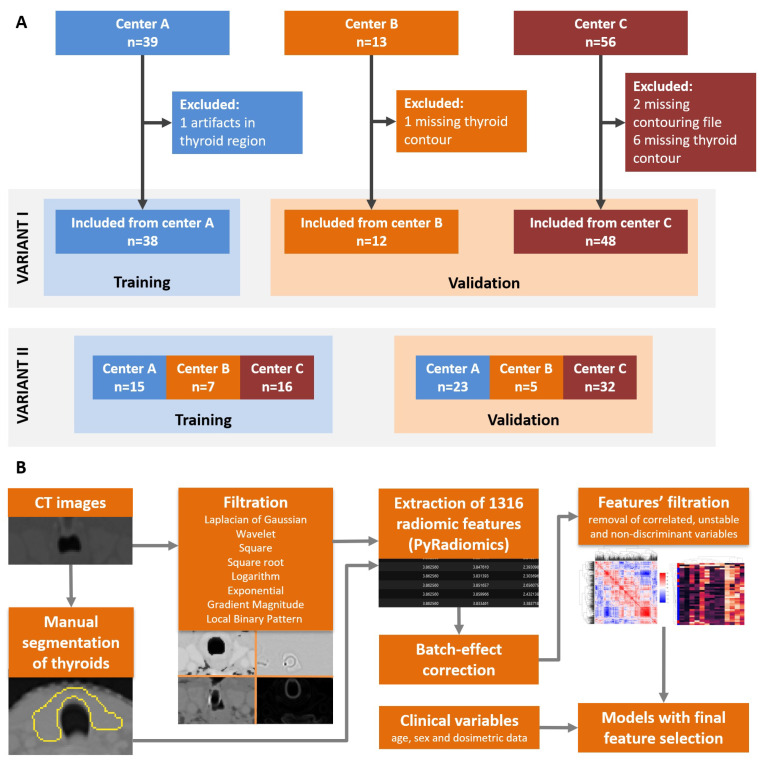
Summary of patients’ selection and data subsets (**A**). Outline of the study from image segmentation to NTPC model derivation (**B**).

**Figure 2 cancers-13-05584-f002:**
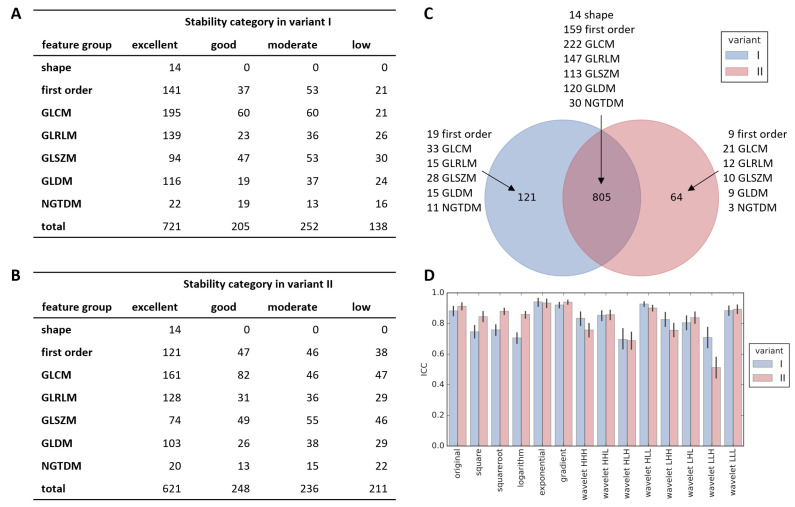
Results of feature stability assessment. Number of features in each group classified to different stability categories for variant I (**A**) and II (**B**). Overlap of features with at least good stability between variant I and II (**C**). Average stability of all radiomic features with respect to applied image filter (**D**, shown mean with 95% confidence interval).

**Figure 3 cancers-13-05584-f003:**
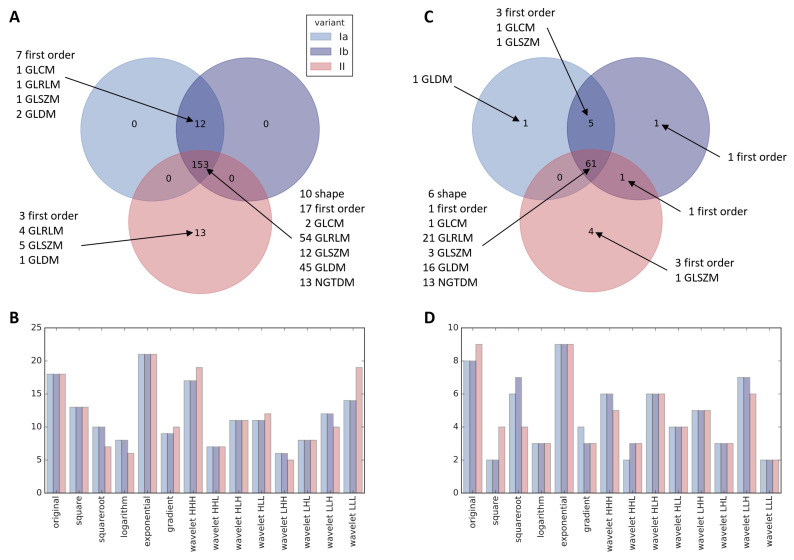
Features with FDR < 0.1 in univariate analysis grouped by radiomics category (**A**) and by image filter (**B**). Features selected for model grouped by radiomics category (**C**) and by image filter (**D**).

**Figure 4 cancers-13-05584-f004:**
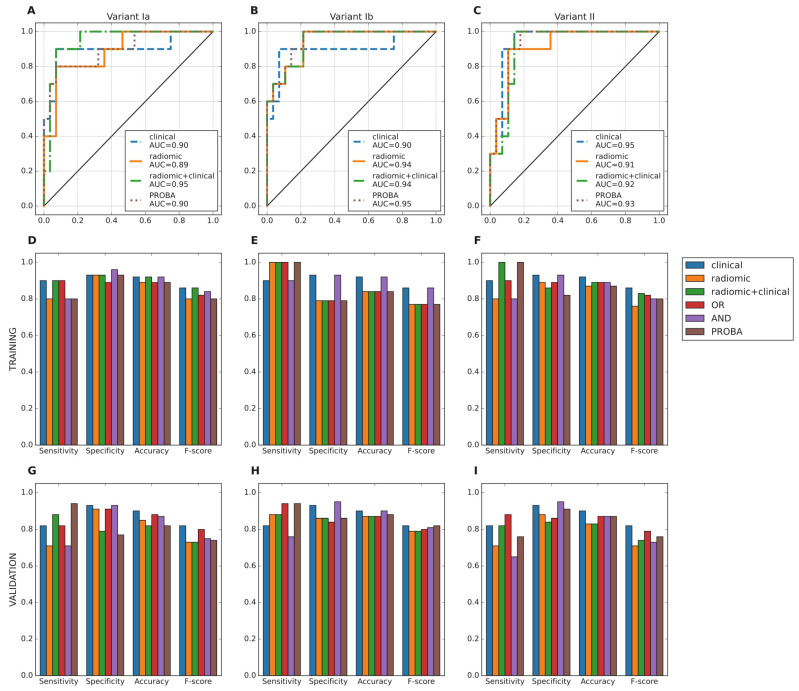
AUC ROC of the selected models (clinical, radiomic, radiomic+clinical) and their ensembles for the training set (**A**–**C**). Comparison of statistical measures for the selected model architectures of the considered feature sets and ensembles of those models: training (**D**–**F**) and validation (**G**–**I**).

**Figure 5 cancers-13-05584-f005:**
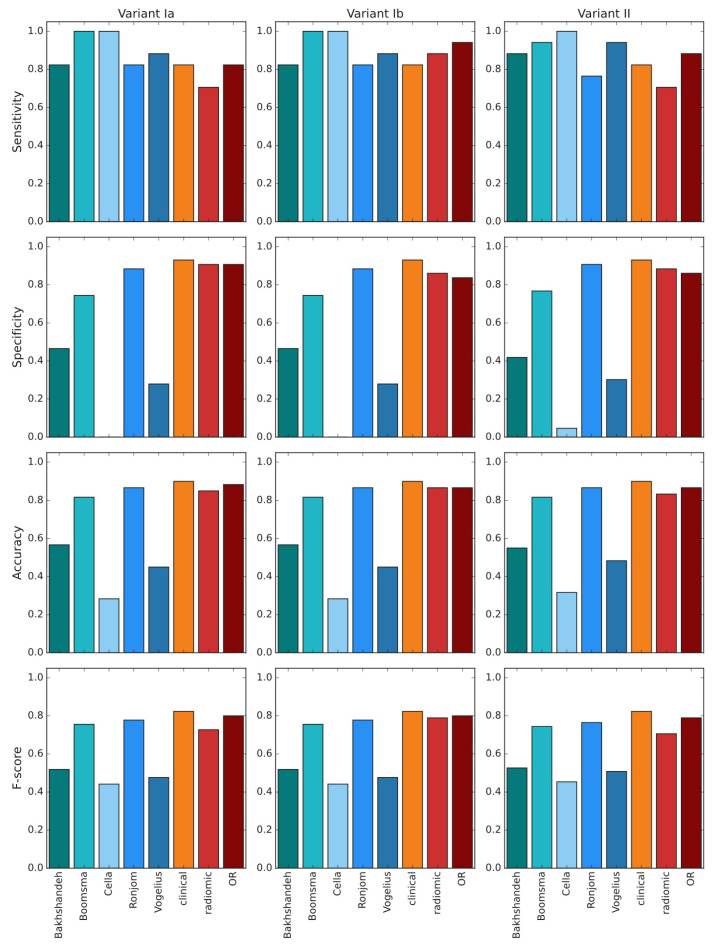
Comparison of quality measures between external NTCP models (cool colors, 5 bars from the left) and our top model selections (warm colors, 3 bars from the right), validation data.

**Figure 6 cancers-13-05584-f006:**
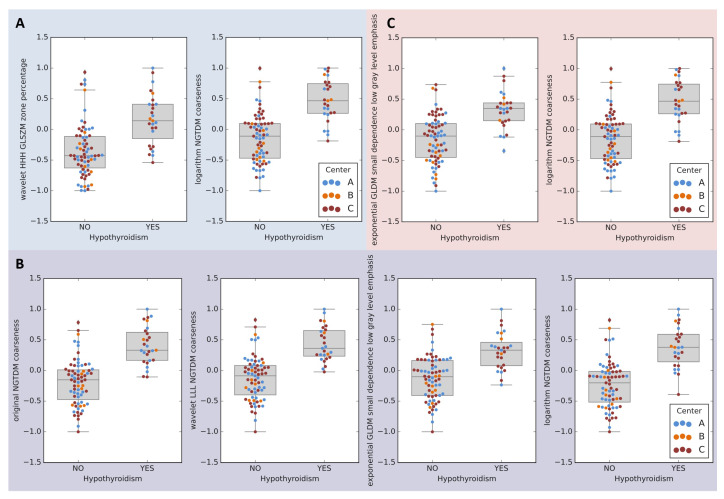
Transformed (normalized and scaled) values of features retained in radiomic models. **A**: variant Ia, **B**: variant: Ib, **C**: variant II.

**Table 1 cancers-13-05584-t001:** Patient, disease and CT images characteristics.

		Center A (n = 38)		Center B (n = 12)		Center C (n = 48)	
RIHT		NO	YES	NO	YES	NO	YES
Sex	Female	7	7	1	1	5	2
	Male	21	3	8	2	29	12
Age	Median	62.0	60.0	61.0	57.0	57.5	58.0
	IQR	57.0–66.2	56.8–61.8	60.0–68.0	55.5–60.5	53.0–62.0	52.2–66.5
Stage	I–II	6	1	2	0	12	1
	III–IV	22	9	7	3	22	13
Mean thyroid dose, D_mean_ (Gy)	Median	54.8	57	55.2	57.3	47.2	52.5
	IQR	51.9–56.3	52.7–59.3	53.5–56.3	56.2–58.4	43.8–49.5	49.7–56.2
Minimal thyroid dose, D_min_ (Gy)	Median	42.5	46.5	43	51.8	30.9	44.3
	IQR	29.1–46.6	43.3–47.5	41.1–48.2	50.1–54.3	24.6–39.0	32.7–46.9
Median thyroid dose, D50 (Gy)	Median	55.0	55.5	54.5	58.5	47.0	52.2
	IQR	53.7–56.3	52.9–58.7	53.9–54.9	57.2–59.0	43.8–50.4	50.5–56.4
Maximal thyroid dose, D_max_ (Gy)	Median	62.5	69.4	61.7	61.6	60.2	65.1
	IQR	57.6–70.1	63.3–71.9	60.8–62.1	59.6–61.8	52.6–68.0	53.6–72.2
Thyroid volume (mL)	Median	21.7	11.8	29	12.6	19.1	10.6
	IQR	19.0–32.9	7.7–13.9	21.7–37.4	10.6–14.0	14.6–27.6	8.3–13.3
Baseline fT4 (pg/mL)	Median	6.5	6.1	9.3	8.2	7.2	8.1
	IQR	5.3–7.4	5.1–7.6	8.0–10.1	7.7–10.7	6.3–8.4	7.9–9.9
Baseline TSH (mIU/L)	Median	0.5	1.3	0.7	0.4	0.7	1.1
	IQR	0.3–0.8	0.8–1.7	0.6–1.2	0.4–0.7	0.5–1.5	0.6–1.2
Mean pituitary dose (Gy)	Median	4.0	3.8	4.0	3.8	3.8	3.7
	IQR	3.0–4.5	3.0–5.3	3.2–4.8	3.6–3.8	3.0–4.4	3.0–4.8
Time to follow–up (months)	Median	29.5	15	22	13	38	19
	IQR	26.0–37.2	14.0–15.8	21.0–24.0	12.0–13.5	31.2–41.0	16.0–21.0
Pixel spacing (mm^2^)	0.98 × 0.98	25	9	0	0	26	11
	1.07 × 1.07	1	0	0	0	3	3
	1.09 × 1.09	0	0	0	0	1	0
	1.11 × 1.11	0	0	0	0	1	0
	1.13 × 1.13	0	0	0	0	1	0
	1.17 × 1.17	0	0	0	0	1	0
	1.27 × 1.27	1	0	9	3	1	0
	1.56 × 1.56	1	1	0	0	0	0
Slice thickness (mm)	1.5	1	0	0	0	0	0
	2	1	1	0	0	2	1
	2.5	0	0	9	3	0	0
	3	24	9	0	0	26	9
	4	0	0	0	0	6	4
	5	2	0	0	0	0	0

**Table 2 cancers-13-05584-t002:** Model architectures and features selected for each of the variants and feature sets.

	VARIANT Ia		VARIANT Ib		VARIANT II	
	Model	Features	Model	Features	Model	Features
clinical (same for Ia and Ib)	GPC	D_mean_ D50 V_thyroid_	GPC	D_mean_ D50 V_thyroid_	GPC	D_mean_ D50 V_thyroid_
radiomic	LR_E_	wavelet HHH GLSZM zone percentage logarithm NGTDM coarseness	MLP_4_	original NGTDM coarseness wavelet LLL NGTDM coarseness exponential GLDM small dependence low gray level emphasis logarithm NGTDM coarseness	MLP_4_	exponential GLDM small dependence low gray level emphasis logarithm NGTDM coarseness
clinical+radiomic	MLP_2_	sex original shape least axis length exponential GLRLM run percentage exponential GLDM small dependence low gray level emphasis logarithm NGTDM coarseness	MLP_4_	original NGTDM coarseness wavelet LLL NGTDM coarseness exponential GLDM small dependence low gray level emphasis logarithm NGTDM coarseness	MLP_2_	sex original shape least axis length exponential GLRLM run percentage exponential GLDM small dependence low gray level emphasis logarithm NGTDM coarseness

GPC–Gaussian process classifier, LR_E_–logistic regression with equal cases weights, MLP_n_–MLP network with n neurons in single hidden layer.

**Table 3 cancers-13-05584-t003:** Comparison between best models with and without radiomic features.

	VARIANT Ia		VARIANT Ib		VARIANT II	
Model	AUC ± SE	*p*	AUC ± SE	*p*	AUC ± SE	*p*
clinical	0.90 ± 0.07	-	0.90 ± 0.07	-	0.95 ± 0.05	-
radiomic	0.89 ± 0.07	0.9196	0.94 ± 0.05	0.6471	0.91 ± 0.07	0.6263
radiomic+clinical	0.95 ± 0.05	0.5549	0.94 ± 0.05	0.6471	0.92 ± 0.06	0.8286
PROBA	0.90 ± 0.07	1.0000	0.95 ± 0.05	0.5549	0.93 ± 0.06	0.7940

*p*-values for comparison of each model with radiomic features vs clinical model.

## Data Availability

Raw, anonymized clinical data and values of calculated radiomic features were included in Appendix A.

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
