# Peer review of "Prediction of Radiation-Induced Hypothyroidism Using Radiomic Data Analysis Does Not Show Superiority over Standard Normal Tissue Complication Models"

_cancers, 2021, doi:10.3390/cancers13215584_

Round 1

Reviewer 1 Report

Very nice paper, high quality, clinically significant. The Radiomic approach is well presented and graphically accurate. The clinical message is relevant because Radiomica analysis did not show any benefit over standard evaluation with a non-Radiomic approach. 

I suggest to be more clear in the last phrases of the abstract to high-light this message.

The conclusions should be written with more clarity because the present form is a little bit "foggy".

Author Response

Thank you for your review of our manuscript and the appreciation of our work. As you suggested, we rephrased the last few sentences of the abstract, highlighting potential utility of radiomic models for RIHT prediction. The conclusion has also been rephrased to make it clearer.

Reviewer 2 Report

The manuscript investigates the prediction of RIHT through models based on clinical, radiomics features, or a combination thereof. 

The main limitation is the number of RITH (i.e., 27 cases) in 98 patients. The limited number of RITH should be discussed considering the possible overfitting of models (reported in Table 2) and the number of identified predictors. 

More details on the resampling approach should be reported in the text.

Lines 131-135: Authors should clarify the reason for scaling two times the raw features in the range [-1,1]

The ICC values were obtained by comparing the 100 different transformation masks. Were all the different transformation masks necessary for obtaining the final models? This issue should be clarified considering the limited dataset. 

Table 3 results for radiomics and clinical for the variant II group should be checked or discussed considering that this group includes the largest number of cases (15/29 cases).  

Author Response

Thank you for thorough review of our manuscript and suggestions for improvements. Please find below answers to the points you raised in your review.

  1. The limited number of RITH should be discussed considering the possible overfitting of models (reported in Table 2) and the number of identified predictors.

We have added this point to the Discussion as a limitation due to a small sample size. The multicenter-approach mitigates the risk of overfitting somewhat, although obviously larger validation would be mandated. Nevertheless, given the minute difference in performance between radiomic and NTCP models, a study aiming at the confirmation of superiority of either would be unfeasibly large.

  1. More details on the resampling approach should be reported in the text.

We added details about image and mask resampling to the text. We used default interpolators and the only setting we changed was pad distance for image and mask cropping to ROI bounding box that was increased from 5 to 10 voxels. Previously, we omitted this information before because we applied majorly default settings recommended by PyRadiomics authors, but we agree that it could have been unclear for the reader.

  1. Lines 131-135: Authors should clarify the reason for scaling two times the raw features in the range [-1,1]

The reason for double scaling has been added to revised version of manuscript. The second scaling is a typical procedure often used before machine learning models are trained. It limits the dynamic range of feature values and facilitates calculation of model’s coefficients by optimization algorithms. First scaling was added to avoid exponentiation of numbers with high absolute values that were observed in some radiomic features, e.g. energy sometimes reached values at the order of 10x109 that can cause computational issues during exponentiation.

  1. The ICC values were obtained by comparing the 100 different transformation masks. Were all the different transformation masks necessary for obtaining the final models? This issue should be clarified considering the limited dataset.

Transformed thyroid masks were used only for stability assessment. Artificial image transformations were treated as simulation of minor inter-specialist differences in contouring and our purpose was to exclude features susceptible to such variations. As it is difficult to determine which transformation reflects inter-specialist differences in the best way, we decided to analyze 100 transforms with randomly selected parameters. At the stage of model derivation, however, features were calculated using only original masks, manually drawn by radiotherapists. We have added an explanation of the role of these transforms to the manuscript.

  1. Table 3 results for radiomics and clinical for the variant II group should be checked or discussed considering that this group includes the largest number of cases (15/29 cases).

We checked results from Table 3 and indeed we had an error in code calculating scores for radiomic+clinical models. Wrong model (other than the actual best one) has been taken for calculations that resulted in seemingly poor performance in Variant II and also slightly altered scores in Variant Ia and Ib. Thank you very much for directing our attention to this issue. We have corrected the error and now results are consistent with radiomic and clinical models.

Variant II training set contained exactly the same number of 38 patients as center A data set (training set for variant I), including 10 cases of RITH that also matches number of such cases in center A data set. We clarified this issue in the text where variants of analysis and data set splitting are discussed.

Reviewer 3 Report

My main suggestion is that the authors should try to explain their experimental findings in pathophysiological terms, based on the existing knowledge on the study topic and on known and/or hypothesized mechanisms of radiation-induced hypothyroidism and their potential correlation with imaging and radiomics data. Such explanation should ideally be given in the Discussion section of the manuscript.

Author Response

Thank you for your review of our manuscript. As you suggested we have added an explanation on potential links between radiomics and pathophysiology; it is now the third paragraph of the discussion. Our main assumption is that differences in thyroid appearance in CT (texture, intensity) result majorly from variation in thyroglobulin-bound iodine accumulation that on its own is related to thyroid function and possibly it’s the organ’s ability to withstand or regenerate after harsh conditions. Such experimental mechanistic investigations are however difficult if not impossible to pursue in actual clinical scenarios like radiotherapy of the OPC. It is, therefore difficult to pinpoint specific pathophysiological  variables to  particular radiomic features based on solely observational results, as they often represent characteristics that are not visible to the naked eye and their intuitive understanding and description in natural language terms are very difficult.

Round 2

Reviewer 2 Report

the manuscript has been improved as suggested